# Self-Supervised MultiModal Versatile Networks

**Jean-Baptiste Alayrac**[1*]    **Adrià Recasens**[1*]    **Rosalia Schneider**[1*]    **Relja Arandjelović**[1*]

**Jason Ramapuram**[2,3†]    **Jeffrey De Fauw**[1]    **Lucas Smaira**[1]    **Sander Dieleman**[1]

**Andrew Zisserman**[1,4]

[1]DeepMind    [2]Faculty of Science, Computer Science Dept., University of Geneva, HES-SO
[3]Geneva School of Business Admin. (DMML Group) [4]VGG, Dept. of Eng. Science, University of Oxford
{jalayrac, arecasens, rgschneider, relja}@google.com
[‡]https://github.com/deepmind/deepmind-research/tree/master/mmv

## Abstract

Videos are a rich source of multi-modal supervision. In this work, we learn representations using self-supervision by leveraging three modalities naturally present in videos: visual, audio and language streams. To this end, we introduce the notion of a *multimodal versatile network* – a network that can ingest multiple modalities and whose representations enable downstream tasks in multiple modalities. In particular, we explore how best to combine the modalities, such that fine-grained representations of the visual and audio modalities can be maintained, whilst also integrating text into a common embedding. Driven by versatility, we also introduce a novel process of *deflation*, so that the networks can be effortlessly applied to the visual data in the form of *video* or a *static image*. We demonstrate how such networks trained on large collections of unlabelled video data can be applied on *video*, *video-text*, *image* and *audio* tasks. Equipped with these representations, we obtain state-of-the-art performance on multiple challenging benchmarks including UCF101, HMDB51, Kinetics600, AudioSet and ESC-50 when compared to previous self-supervised work. Our models are publicly available[‡].

## 1   Introduction

Our experience of the world is multimodal. From as far back as the crib, we perceive through multi-sensory systems, for instance we *watch* the flames dancing in the fireplace, we *hear* the sound of the crackling wood, as well as *feel* the heat coming off. Through this multimodal synchronous perception, we learn to draw useful connections between modalities [73] which, in turn, enables us to form good representations of the world. Later, comes *language* that allows us to communicate this fine-grained multimodal experience using higher-level abstract concepts.

Our objective is to learn representations from such multimodal experience in a self-supervised manner without resorting to any specific manual annotation. The modalities considered are the three that are easily available from large collections of unlabelled videos: visual, audio and language (obtained from narrations) streams. In this, we seek to learn a *multimodal versatile network*, defined as a network that has the following four properties: **(i)** it should be able to take as input any of the three modalities; **(ii)** it should respect the specificity of modalities, in particular the fact that the audio and visual modalities are much more fine-grained than language; **(iii)** it should enable the different

---

[*]Equal contribution.
[†]Work done during an internship at DeepMind.

modalities to be easily compared even when they are never seen together during training; and finally **(iv)** it should be efficiently applicable to visual data coming in the form of dynamic videos or static images.

The question is how to design a network that respects these four principles? We choose a design that embeds each modality into a vector space such that similarity between modalities is obtained via simple dot products. Each modality is processed by a backbone network adapted to the nature of the signal, and a *modality embedding graph* is constructed such that the visual and audio embeddings are fine-grained, whilst the textual embedding is semantically coarse-grained. This strategy is based on the observation that the visual and audio spaces are fine-grained (there are many visual or sounds of *guitars* that might be really different to each other) while the textual domain is more coarse as its goal is to abstract away details (*e.g.* a single "*guitar*" word). The network is then trained from scratch via self-supervised contrastive learning on a large set of unlabelled videos.

To quantitatively evaluate our learned MultiModal Versatile (MMV) networks, we measure their performance on multiple downstream tasks, and in this way assess various properties of the representation of videos and images: *verb learning* (action classification on HMBD51, UCF101 and Kinetics600); *noun learning* (image classification on PASCAL VOC and ImageNet); *joint text and visual representation* (YouCook2, MSRVTT); and *audio representation* (sound classification on ESC-50 and AudioSet). The proposed MMV achieves state-of-the-art performance for self-supervised approaches on these benchmarks, and reduces the gap to the state-of-the-art performance for supervised approaches.

**Contributions.** Our contributions are the following: **(a)** we investigate different modality embedding graphs for MMV, and propose a simple yet effective self-supervised training strategy for multimodal representation of audio, visual and language streams; **(b)** we introduce a deflation approach so that the MMV video network can efficiently ingest a static image; and **(c)** we demonstrate the superiority of the learned representations on multiple *image*, *video*, *audio* and *video-text* downstream tasks.

## 2    Related work

**Self-supervised learning from single modality.**   Self-supervised methods design pretext tasks that require no manual annotation but facilitate learning of useful representations of the data. A variety of pretext tasks have been developed for vision (*i.e.* single modality), such as predicting the relative position of patches [15, 56], colorization [92], predicting orientation [24] or invariance to transformation [17, 33]. In videos, works have also leveraged the temporal dimension [20, 43, 53, 88]. Recently, methods that maximise the similarity between multiple views (augmented versions) of the same image via contrastive losses [9, 13, 29, 30, 31, 57] stand out due to impressive results on the ImageNet benchmark; we draw inspiration from them (*e.g.* use a contrastive loss and non-linear projection heads [13]). However, details of view generation are crucial and require careful design [78]. In contrast, we argue that using multiple modalities as different views is simpler and more natural [77].

**Vision and language.**  WSABIE [83] and DeVise [21] introduced the idea of embedding text and image in the same space. This allows semantic similarity to be measured by a dot product in a vector space and enables fast and efficient large scale search across modalities [34]. This idea is at the core of our versatile networks. With larger datasets [45, 66, 68, 89, 93], many works have profited from learning such a joint visual-textual space [14, 16, 26, 27, 38, 50, 54, 60, 65, 81, 82, 84, 85, 90]. Recently, instructional videos became a popular source of video and language data [3, 48, 72, 91] due to not requiring careful manual annotation, *e.g.* by using Automatic Speech Recognition (ASR) to generate text from narrations. We build on top of [49, 75, 76] who learn good representations from such narrated material, but consider learning representations using audio as well.

**Vision and audio.**  Cross-modal teacher-student methods [7, 59] exploit the temporal co-occurrence between visual and audio modalities in a video to learn good representations. Taking this idea into the self-supervised domain [5], multiple works use a pretext task of predicting whether visual and audio signals come from the same video [5, 6, 41, 55, 58, 69]. Recent developments such as XDC [4], who employ cross-modality clustering, or Evolving Losses [64], where m any single- and multi-modal pretext tasks are used, demonstrate an impressive ability to learn good representations in both modalities. We propose a simpler method that achieves better performance, and consider the text modality as well.

**Vision, audio and language.** Using audio, vision and language to learn representations has also been explored in past work [8, 28, 35, 47, 80]. In particular, Harwath *et al.* [28] use a dataset of images and audio descriptions to associate spoken words and their visual representation. Similarly to us, Aytar *et al.* [8] train a cross-modal network with image, audio and text modalities. One major difference is that they rely on curated annotated datasets, while our approach requires no annotation.

**From video to image.** We reverse the usual route of going from an image network to a video network by *inflation* [12]. Historically, this was the usual route [25] as labels were more readily available for images, *e.g.* ImageNet, than for videos. However, our perception of the world is actually dynamic, a time series of images, and learning first from videos is more natural. Similarly to [16], we enable our network to ingest both dynamic video and still images. But instead of having two different pathways and requiring to learn from both images and videos, we propose a simple *deflation* mechanism that enables our network purely trained on videos to be directly adapted to still images.

## 3 Approach

We are given a set of *unlabelled* videos containing different modalities. For example, a video may contain an RGB stream (*e.g.* a set of frames depicting a dog), an audio track (*e.g.* the sound of that same dog barking) and some linguistic narrations (*e.g.* coming from a person providing verbal instructions). We follow previous work [49, 51] and obtain language as text by using off-the-shelf Automatic Speech Recognition (ASR) on the audio, leaving the removal of this dependency to future work. Equipped with this, our goal is to learn a model that has the *versatile* properties described in Section 1. We do so by introducing a bespoke multimodal architecture and optimize its parameters via self-supervised learning. In details, we use the temporal co-occurrence between the modalities to define the self-supervised proxy task and enforce it with a multi-modal pairwise contrastive loss.

Formally, a video $x \in \mathcal{X}$ is defined by an instantiation of different modalities $\mathcal{M}$: $x = \{x_m\}, m \in \mathcal{M}$. In this work, we focus on three modalities, namely vision $x_v \in \mathcal{X}_v$, audio $x_a \in \mathcal{X}_a$ and text $x_t \in \mathcal{X}_t$ but the proposed approach could be easily generalized to more modalities. Specifically, $x_v$, $x_a$ and $x_t$ correspond to few-second sequence of RGB frames, 1D audio samples, and discrete word tokens, respectively. Given a training set containing $n$ videos $\{x^i\}_{i=1}^n \in \mathcal{X}^n$, we seek to learn modality specific representations as well as ways to compare streams across modalities. To that end, let $f_m : \mathcal{X}_m \to \mathbb{R}^{d_m}$ be a parametrized modality specific backbone neural network that takes as input an instance $x_m$ from modality $m$ and outputs a representation vector of dimension $d_m$. To compare different modalities together via simple dot products, we embed them into a shared space $\mathcal{S}_s \subset \mathbb{R}^{d_s}$ of dimension $d_s$, where $s$ contains the list of modalities that we embed in the space, *e.g.* $s = va$ for a joint visual-audio space $\mathcal{S}_{va}$, or $s = vat$ for a joint visual-audio-text space $\mathcal{S}_{vat}$. A modality specific representation $f_m(x_m)$ is embedded into a space $\mathcal{S}_s$ via a projection head $g_{m \to s} : \mathbb{R}^{d_m} \to \mathbb{R}^{d_s}$. We denote by $z_{m,s} = g_{m \to s}(f_m(x_m))$ the vector representing the input modality $x_m$ in the space $\mathcal{S}_s$.

Section 3.1 explores various model design choices for the MMV networks, which induce different structures of modality spaces $\mathcal{S}_s$. It also presents the self-supervised losses that enforce the different modalities to align in the common spaces. In Section 3.2, we explain how to simply adapt models that have been trained on sequences of RGB frames to operate on single frames.

### 3.1 MMV: MultiModal Versatile Networks

Recall our goal is to be able to embed different modalities into a vector space where semantic comparisons can be made by simple dot products. Since there are three modalities, multiple configurations of modality spaces with different inter-relations, which we call *modality embedding graphs*, can be envisaged. An important note is that since the text modality is directly obtained from the audio track using ASR, we do not construct the audio-text space nor the loss that puts them in alignment *explicitly*. This is because our goal is not to learn ASR but instead to associate a word, *e.g.* "car", with the sound associated with that entity, *e.g.* the sound produced by the engine. However, we hypothesize that the model can learn this desired link *implicitly* thanks to the common visual ground. We consider three options for the *modality embedding graphs*, illustrated in Figure 1 and detailed next.

*Option I: Shared space.* This is the simplest model where all modalities are embedded into a single shared vector space $\mathcal{S}_{vat} \subset \mathbb{R}^{d_s}$, and in which direct comparisons can be made between modalities (Figure 1a). For example, starting from a visual vector $f_v(x_v)$, a single projection head is applied

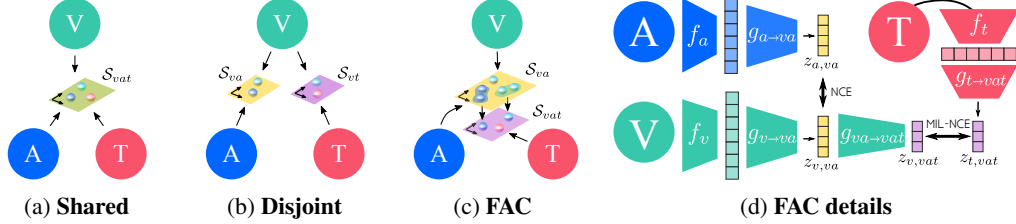

|           |           |           |              |
|:---------:|:---------:|:---------:|:------------:|
| (a) **Shared** | (b) **Disjoint** | (c) **FAC** | (d) **FAC details** |

Figure 1: (a)-(c) Modality Embedding Graphs, (d) Projection heads and losses for the FAC graph. V=Vision, A=Audio, T=Text.

to obtain the embedding $z_{v,vat}$ used to compare to the audio and the text modalities. This strategy has the advantage that it is easy to navigate between modalities since they all live in the same space (property **(iii)**). However, the model implicitly assumes that all modalities have equal granularity and hence does not respect their specificities (lack of property **(ii)**).

*Option II: Disjoint spaces.* Another natural option is to have different visual-audio ($\mathcal{S}_{va}$) and visual-text ($\mathcal{S}_{vt}$) spaces, as illustrated in Figure 1b. For example, starting from the visual representation $f_v(x_v)$, there are two distinct projection heads mapping to the $\mathcal{S}_{va}$ and the $\mathcal{S}_{vt}$ domains, *i.e.* $z_{v,va} \neq z_{v,vt}$. While the *disjoint spaces* approach enables the specificity of different modality pairs (property **(ii)**), it does not allow easy navigation between the embedding spaces (lack of property **(iii)**), for example, text to audio retrieval (*e.g.* "car" to "engine sound") is not possible.

*Option III: Fine and coarse spaces (FAC).* In the introduction, we argue that the visual and the audio domains are different from the language domain in terms of their granularities. Inspired by this intuition, we propose to learn two embedding spaces: vision and audio are compared in the fine-grained space ($\mathcal{S}_{va}$), while text is compared with vision and audio in the lower dimensional coarse-grained space ($\mathcal{S}_{vat}$). Crucially, vectors in $\mathcal{S}_{va}$ can be embedded into $\mathcal{S}_{vat}$ via a simple fine-to-coarse projection $g_{va \to vat}$, as illustrated in Figure 1c. For example, to compare vision to audio, the visual representation is projected into the fine-grained space $\mathcal{S}_{va}$ via $g_{v \to va}$. To compare vision to text, vision is embedded into the coarse-grained space $\mathcal{S}_{vat}$ via projection $g_{v \to vat}$ which decomposes as $g_{va \to vat} \circ g_{v \to va}$; this can be seen as first projecting the vision into the fine-grained space $\mathcal{S}_{va}$ via $g_{v \to va}$, followed by projecting the fine- into the coarse-grained space by $g_{va \to vat}$ (see Figure 1d). Note that even though we do not align audio and text during training (as mentioned before, this is to not learn ASR), the imposed modality embedding graph enables audio-text comparison because audio can still be projected into the coarse-grained space $\mathcal{S}_{vat}$ via $g_{va \to vat} \circ g_{a \to va}$. This strategy covers the three relevant properties of the MMV network – as opposed to the *shared space* solution, it models the text differently from the vision and the audio (property **(ii)**), and, in contrast to the *disjoint spaces* approach, it enables easy navigation across modalities (property **(iii)**).

**Multimodal contrastive loss.** Given the previously described embedding graphs joining the three different modalities, the question remains how to actually train the backbones and the projection heads. We wish to do so without resorting to any form of manual annotations in order to leverage large amounts of readily available videos on the internet. Instead, inspired by [5, 49], we construct self-supervised tasks which aim to align pairs of modalities: vision-audio or vision-text, but not audio-text as explained earlier. Concretely, *positive* training pairs across two modalities are constructed by sampling the two streams from the same location of a video. Conversely, *negative* training pairs are created by sampling streams from different videos. In practice, a minibatch of $N$ video samples is formed, which induces $N$ positive and $N^2 - N$ negative pairs. Given these positive and negative training pairs, we use a contrastive loss [30, 49, 57] to make the positive pairs similar and negative pairs dissimilar in their corresponding joint embedding space. The only difference between losses used with different *embedding graph* designs is the choice of spaces where the dot products are computed; next we give the loss for *FAC* and provide the *shared* and *disjoint* losses the extended version [1]. Formally, given a video $x$, we minimize the multimodal contrastive loss:

$$\mathcal{L}(x) = \lambda_{va}\text{NCE}(x_v, x_a) + \lambda_{vt}\text{MIL-NCE}(x_v, x_t), \qquad (1)$$

where $\lambda_{mm'}$ corresponds to the weight for the modality pair $m$ and $m'$. The component corresponding to the visual-audio pair is the following NCE loss (for FAC):

$$\text{NCE}(x_v, x_a) = -\log\left(\frac{\exp(z_{v,va}^\top z_{a,va}/\tau)}{\exp(z_{v,va}^\top z_{a,va}/\tau) + \sum_{z' \sim \mathcal{N}(x)} \exp(z'^\top_{v,va} z'_{a,va}/\tau)}\right), \qquad (2)$$

where $\mathcal{N}(x)$ is a set of negative modality pairs for the video $x$, and $\tau$ is the temperature parameter. For the text, recall that we use narrations automatically obtained from speech. As opposed to the audio that is usually better aligned with its visual source (*e.g.* the sound of the piano is synchronized with the visual of the instrument being played), the correspondence between narrations and what is actually happening in the video is much weaker [49]. To address that issue, we use the MIL-NCE variant from [49] that is tailored to account for this misalignment issue. In short, it considers multiple positive candidate pairs as positives by simply replacing the single term $\exp(z_{v,vat}^{\top} z_{t,vat}/\tau)$ in the standard NCE equation (2) by a sum of scores over positive text candidates: $\sum_{z \in \mathcal{P}(x)} \exp(z_{v,vat}^{\top} z_{t,vat}/\tau)$. As in [49], the set of potential positives $\mathcal{P}(x)$ is formed from temporally close narrations.

**Missing modalities.** Some videos do not have all modalities, for example not all videos contain narration. In that case, we simply discard the corresponding loss component in (1) and upweight the remaining examples of the same modality pair in the batch in order for the total loss weight to remain constant.

### 3.2 Video to image network deflation

To comply with the property **(iv)** of the *multimodal versatile* network, we introduce a *network deflation* operation to transform a video network into a network that can ingest a single image. The deflated network can be evaluated on *image* downstream tasks while training on videos, and is more efficient than the standard trick of assembling a static video by repeating the image in time.

Ideally we would wish for video-image equivalence: that the output of the deflated video network on a single image is identical to that obtained by applying the original video network to the single-image static-video. It might be thought that this can simply be achieved by deflating the network over the temporal dimension. In the two types of video networks considered here, this deflation corresponds to the following operations: for 3D convolutional based networks [12, 87], summing the 3D spatio-temporal filters over the temporal dimension to obtain 2D filters; for TSM networks [44], turning off the channel shifting which results in a standard residual architecture (*e.g.* ResNet50) for images.

However, due to zero-padding these operations do not achieve the desired equivalence – since filters whose receptive field overlap the clip boundary receive zeros in the single-image static-video, and this is not taken into account by the simple deflation operation above. Note, the padding used in the spatial domain is not a problem, as the spatial padding applies equivalently for both video frames and single images. To take account of the zero-padding, we learn new parameters $\gamma$ and $\beta$ for the batch normalization layers to correct for this boundary effect on the filter outputs, and *approximate* the equivalence we seek. In detail, the $\gamma$ and $\beta$ parameters are trained to minimize the $L_1$ loss between the output of the original video network when presented with single-image static-videos, and the output of the deflated network for the same images; all parameters are frozen apart from $\gamma$'s and $\beta$'s of the deflated network. Note that this process only requires images without the need for annotations.

## 4 Experiments

In this section we evaluate the performance of the networks on a wide range of downstream tasks. We start by describing the experimental protocol and the datasets used for self-supervised pretraining and downstream evaluations (Section 4.1), followed by exploring various design choices (Section 4.2). Based on this study, we train final models at a larger scale to compare them to the state-of-the-art (Section 4.3). Finally, we apply the trained video networks on still image tasks (Section 4.4).

### 4.1 Experimental setup, datasets and downstream tasks

**Network architectures, hyperparameters and optimization.** For video we explore using S3D-G [87] ($d_v = 1024$), and TSM [44] with a ResNet50 backbone ($d_v = 2048$) or a ResNet50x2 backbone (ResNet50 with all channels doubled [39], $d_v = 4096$). We apply temporal and spatial average pooling at the last layer of the backbone (before the usual classification layer) to obtain a single vector $f_v(x_v)$. During training, 32 (16 for the exploration design) frames are sampled at 10 fps and $200 \times 200$ crops are used (frames are resized so that the minimum side is 224). We use the following standard augmentation during training: random crop, horizontal flipping, temporal sampling and scale jittering, and color augmentation (details in the extended version [1]). Audio

is represented as log MEL spectrogram with 80 bins and processed with ResNet50 and is sampled in sync with the frames. Spatial pooling is applied to obtain $f_a(x_a)$ of dimension $d_a = 2048$. For the final audio evaluation (Section 4.3), the network ingests 2 seconds of audio for fair comparison to [4, 41], otherwise we use the same duration as the input video clip. Following [49], text is processed by removing stop words, retaining a maximum or padding to 16 words, then extracting 300-dimensional Google News pre-trained word2vec [52] and finally applying a linear layer to independently map the word inputs to 2048 dimension followed by a max pooling layer over the 16 words ($d_t = 2048$). The dimension of the shared subspaces is 512, except for the Fine And Coarse (FAC) design where we use 512 dimensions for $\mathcal{S}_{va}$ (fine) and 256 for $\mathcal{S}_{vat}$ (coarse). More details about architecture are provided in the extended version [1]. As done in [13], we normalize vectors prior to computing their dot products in the NCE and MIL-NCE losses and use a temperature of $\tau = 0.07$ in the softmax as in [29, 62, 86]. When training with all three modalities on HowTo100M, we observe that a larger weight on the Vision-Text loss is beneficial since text is more prominent. However, when training on HowTo100M+AudioSet, equal loss weights worked best because the audio from AudioSet is more informative. Therefore, a 10:1 loss weight ratio is used when training on HowTo100M and 1:1 for HowTo100M+AudioSet. Finally, all networks are trained from scratch using Adam [37] with an initial learning rate of $0.002$, $5K$ steps of warm up and a half-period cosine schedule [46].

**Training datasets.** We use the HowTo100M [51] and/or the train split of AudioSet [22] datasets for self-supervised training. The HowTo100M dataset contains more than 100 millions narrated video clips coming from 1 million unique videos where the audio narration is transcribed into text using ASR. We follow the same processing as described in [49] for creating positive and negative pairs for our contrastive based loss. AudioSet consists of 10 seconds clips coming from 2 million different internet videos. The dataset contains a variety of audio tracks such as musical instruments, animals or mechanical sounds, since it was built for audio classification, but we discard the labels for self-supervised training. Due to the dataset nature, text is considered a *missing modality* for AudioSet.

**Downstream tasks.** The trained networks are evaluated on various downstream tasks that aim to capture different aspects of the learned representations. For *action classification*, we evaluate the visual representation on the UCF101 [74] (13K videos and 101 action classes) and the HMDB51 [42] (7K videos and 51 classes) benchmarks. Two settings are explored – *frozen* where we simply learn a linear classifier on top of the pretrained $f_v(x_v)$ vector, and a *finetune* setting where the full visual model $f_v$ is finetuned. We also propose an additional large scale downstream task by evaluating the performance on Kinetics600 [10] (30K evaluation clips with 600 classes) in the *frozen* setting. To evaluate the quality of the *audio representation*, we use the ESC-50 [63] (2K audio clips with 50 classes) and AudioSet [22] (20K eval audio clips with 527 classes) classification task using the *frozen* setting on the features produced by the last convolution of the audio backbone network. We report mAP on AudioSet and the top-1 accuracy for ESC-50. Some classification datasets have official splits (3 for UCF101/HMDB51 and 5 for ESC-50). As per standard, split#1 serves as the validation set and is therefore used for ablations (Section 4.2), and the average accuracy over all splits is reported when comparing to the state-of-the-art (Section 4.3). The quality of our *text-video* representation is evaluated on *zero-shot* text-to-video retrieval on the MSRVTT [89] (1K videos) and YouCook2 [93] (3190 videos at the time of publication) benchmarks, by following the evaluation protocol described in [51] and reporting the recall at 10 (R@10) (and other retrieval metrics in the extended version [1]). Finally, to evaluate how well our video representation transfers to *image* tasks we use the PASCAL VOC 2007 [18] and ImageNet [70] classification tasks. For the *image* tasks, the *frozen* setting on the *deflated* version of $f_v$ is used, and, as per standard, we report the mAP on PASCAL and the top-1 and top-5 accuracies on ImageNet. Full details are given in the extended version [1].

## 4.2 Design explorations

We here summarize the effects of various design choices of our method. To facilitate running a large number of experiments, we use the S3D-G [87] network as the video backbone, with 16 frames per video clip, a total batch size of $512$ and 500K training steps (20 hours training on 16 Cloud TPUs). Unless otherwise stated, linear projection heads are used for all modalities, and the networks are trained on HowTo100M. To minimize the amount of hyper-parameter tuning, for UCF101, HMDB51 and ESC-50 we use only the *frozen* setting and report top-1 accuracy on the split#1. We also report R@10 for YC2 (YR10) and MSRVTT (MR10) under the zero-shot setting. Full details, including all quantitative results, are given in the extended version [1].

Table 1: **Design explorations for multiple modalities (HT=HowTo100M, AS=AudioSet).** The video networks use non-linear projection heads.

(a) **Benefits of multiple modalities on HT**

| Modalities | UCF | HMDB | YC2 | MSRVTT | ESC-50 |
|---|---|---|---|---|---|
| VT | 82.7 | 55.9 | **33.6** | 27.5 | / |
| VA | 75.5 | 51.6 | / | / | **79.0** |
| VAT (FAC) | **84.7** | **57.3** | 32.2 | **28.6** | 78.7 |

(b) **VAT: modality merging strategies on HT+AS**

| Strategy | UCF | HMDB | YC2 | MSRVTT | ESC-50 |
|---|---|---|---|---|---|
| Shared | 84.7 | 60.2 | 20.8 | 22.4 | **88.5** |
| Disjoint | 85.1 | 59.3 | **25.0** | 22.5 | 87.0 |
| FAC | **86.2** | **62.5** | 23.8 | **23.5** | 88.0 |

**Pairs of modalities.** We here summarize the main findings from experiments that consider learning from two modalities – Vision and Text, or Vision and Audio – as this setup makes it easy to isolate the effects of different components and discover the best building blocks to be used in the three-modality setting. For the *video backbones*, we observe that TSM ResNet50 always beats S3D-G for downstream tasks that involve vision. For Vision and Audio, *contrastive based loss* consistently outperforms logistic loss (used in [5, 41]) by 2% on vision downstream tasks, and is on par for audio. This is in line with findings of recent single-modality self-supervised approaches as well as work in Vision and Text [49] that demonstrate the superiority of NCE based loss compared to its binary classification counterpart. Regarding the *projection heads*, the experiments confirm findings of [13] that adding a non-linear projection head (two layers MLP with BatchNorm and ReLU activations) on top of the visual representations helps (notably for UCF101 and HMDB51). It was not beneficial to have non-linear projection heads for the language and audio branches. We hence keep linear projection heads for audio and text branches and use a non-linear projection head for vision in the rest of the paper. Regarding *data augmentation*, we observe that despite training on large datasets, removing visual augmentation such as color augment or scale jittering slightly decreases performance, hence we keep them for the rest of the paper. Concerning the audio, we add Gaussian noise to the raw signal, with mean 0 and variance $0.01 \times max\ amplitude$, which seems to slightly improve results. Mildly jittering with SpecAugment [61] was not beneficial, and more aggressive augmentations were detrimental; in contrast with the findings of [62] where SpecAugment helped, presumably due to training on a relatively small dataset. Temporal jittering by randomly offsetting the audio with respect to the visual stream by up to 0.8s (half of the training clip length) reduces the performance on visual tasks by 4%, showing that synchronization is an important training signal.

**Combining Vision, Audio and Text.** On HowTo100M, learning with all *three modalities* clearly outperforms networks trained with only pairs of modalities (Table 1a), obtaining significantly better visual representations (UCF101 and HMDB51) and on-par audio representations (ESC-50). The scores are tied on Vision-Text tasks, with the 3-modality network winning on MSRVTT but losing on YC2. These results demonstrate the ability of our network to learn from the complementary training signals coming from the audio and the text modalities. Next we look at the performance of the different *modality merging strategies* on the combination of HowTo100M and AudioSet in Table 1b. First, comparing to Table 1a, we observe that combining AudioSet with HowTo100M improves performance on HMDB51, UCF101 and ESC-50. This confirms again that our networks can leverage the complementary nature of the modalities to learn better representation as well as showcases the advantage of being able to cope with heterogeneous sources of data (AudioSet does not have text). We note a decrease in performance for the video-text benchmarks (MSRVTT and YC2), which can simply be explained by the fact that only a half of the training samples contain text vs. Table 1a (the other half comes from AudioSet which does not have text). As shown in the next section, this can simply be recovered by training for longer. Second, we note that all strategies for merging the three modalities obtain good representations, but the *fine-and-coarse* (FAC) method dominates on UCF101, HMDB51 and MSRVTT, achieves a good result on ESC-50 and is second best on YC2. The result agrees with the intuition that care should be taken to account for the specificity of the different modalities.

### 4.3 Large-scale experiments and comparison to the state-of-the-art

**Final experimental setup.** We use 32 frames per video clip, 500K training steps, and a total batch size of 4096 (S3D-G and TSM-50) or 2048 (TSM-50x2); training TSM-50 takes 3 days on 32 Cloud TPUs. Based on our ablations, the audio and text networks employ a linear projection head, whereas the video network uses a non-linear head. All models use the FAC design when working with the three modalities. Self-supervised training is performed on the combination of HowTo100M and AudioSet datasets with standard augmentation. The full details are in the extended version [1].

**Results.** Table 2 shows our visual and audio representations match or outperform the state-of-the-art on all downstream tasks and evaluation modes (linear or finetuning). Impressively, simple linear classifiers are competitive with some of the best previous work that uses finetuning and set a strong baseline on the large scale Kinetics600 downstream task. We also compare to the *best* externally reported supervised pretraining transfer as a meaningful and strong baseline that self-supervised methods should aim to surpass. Under that challenging comparison, MMV performance on HMDB51 and UCF101 is getting close to the best supervised method that leverage both ImageNet and Kinetics [87], while on ESC-50 it is even better than the best supervised result [71] by 1.7%.

Comparison with previous works on equal grounds is difficult due to the wide range of backbone architectures and training data sources used. Using the same visual backbone (R(2+1)D-18 [79]), training dataset (AudioSet) and modalities (Vision+Audio), we obtain similar performance to XDC [4] and GDT [62] on UCF101, and significantly outperform them on HMDB51. Comparing to best reported results across past works – our smaller TSM-50 model (trained on Vision+Audio+Text) achieves similar performance to GDT [62] while being superior to XDC [4] and ELo [64], despite having significantly fewer parameters and being trained with the same amount or less data; note also that XDC [4] and GDT [62] train on Instagram65M [23] which has been collected specifically to mimic action recognition datasets. The superior performance of the larger TSM-50x2 model demonstrates that large networks can benefit from self-supervised training on vast amounts of data, and that our self-supervised task facilitates this process. This has also been observed in previous work in the image domain [13] and is also confirmed by the better performance of our R(2+1)D-18 backbone versus S3D-G when finetuning on HMDB51 and UCF101.

Comparing to the two-modality case – Vision+Text with S3D-G is a similar setup to [49] and training with three modalities is clearly beneficial. Similarly, FAC also beats training with only Vision+Audio, confirming again the advantage of learning from three modalities instead of two. This is particularly significant on the Kinetics600 downstream task ($+9.2\%$) where the semantic contained in the narrations from HowTo100M about objects or actions may be relevant for the Kinetics classes.

Regarding zero-shot video to text retrieval our MMV S3D-G, TSM-50 and TSM-50x2 respectively obtain a R@10 of 37.2, 41.5 and 45.4 on YouCook2 and 29.3, 31.1 and 31.1 on MSRVTT. As explained in Section 4.2, longer training significantly improves the performance on these two benchmarks when compared to the results reported in Table 1b. We are also not far from the state-of-the-art performance reported in [49] for MSRVTT (32.2) and still below for YouCook2 (51.2). However, Miech *et al.* [49] train 4 times longer on vision-text pairs (same number of total training steps, but $2\times$ larger batches and half of our samples come from AudioSet which has no text). We believe this gap could be further reduced by longer training but leave that for further investigation.

### 4.4 Transfer to image tasks *via* network deflation

**Experimental setup.** The best MMV networks trained in Section 4.3 are deflated and evaluated on image tasks. The deflation (Section 3.2) is trained on 45981 frames of the HowTo100M [51] training set, where the static videos (ingested by the original video network to produce the regression targets for the deflated image network) are 32-frame long to match the video length used during self-supervised training; the Adam optimizer [36] is used with initial learning rate of $10^{-2}$ decayed by a factor 0.1 every 30 epochs for a total of 100 epochs. Results are reported for linear classification on top of the frozen image features $f_v(x_v)$ on the PASCAL VOC 2007 and ImageNet benchmarks. Implementation details are provided in the extended version [1].

**Results.** Table 3 shows that the deflated networks perform almost as well as the original video model applied on input-inflated 32-frame static videos (the difference is only around 1% when comparing the 'def' and 'i-inf' results). However, the deflated model is an order of magnitude more efficient due to processing single images instead of the full 32-frame videos. Naive deflation underperforms severely due to the strong padding effects, proving that our deflation training is necessary. The state-of-the-art self-supervised models trained on images (SimCLR [13]) outperform MMV due to not having to bridge the video-image domain gap and in fact has been trained on ImageNet images – the performance difference is much smaller on PASCAL. Finally, our approach is significantly better than pre-training in a fully supervised manner on Kinetics-700 [11].

Table 2: **Comparison of learnt representations versus the state-of-the-art.** Results are averaged over all splits. The "Mod." column shows which combinations of modalities are used by the methods, possibilities: **V**ision, **A**udio, **T**ext, **F**low. Dataset abbreviations: **A**udio**S**et, **H**ow**T**o100M, **I**nsta**g**ram**65M** [23], **S**ound**Net** [7], 2M videos from **Y**ou**T**ube**8M** [2], **K**inetics**600**; their length in years is given in the "years" column. [†][71] uses a non-linear classifier. We report top-1 accuracy for UCF101, HMDB51, ESC-50, Kinetics600 and mAP for AudioSet.

| Method | $f_v$ (#params) | Train data | years | Mod. | UCF101 | | HMDB51 | | ESC-50 | AS | K600 |
| --- | --- | --- | --- | --- | --- | --- | --- | --- | --- | --- | --- |
| | | | | | Linear | FT | Linear | FT | Linear | MLP | Linear |
| MIL-NCE [49] | I3D (12.1M) | HT | 15 | VT | 83.4 | 89.1 | 54.8 | 59.2 | / | / | |
| MIL-NCE [49] | S3D-G (9.1M) | HT | 15 | VT | 82.7 | 91.3 | 53.1 | 61.0 | / | / | |
| AVTS [41] | MC3 (11.7M) | AS | 1 | VA | | 89.0 | | 61.6 | 80.6 | | |
| AVTS [41] | MC3 (11.7M) | SNet | 1 | VA | | | | | 82.3 | | |
| AA+AV CC [32] | RN-50 (23.5M) | AS | 1 | VA | | | | | | 28.5 | |
| CVRL [67] | R3D50 (33.3M) | K600 | 0.1 | V | | | | | | | 64.1 |
| XDC [4] | R(2+1)D-18 (33.3M) | AS | 1 | VA | | 91.2 | | 61.0 | 84.8 | | |
| XDC [4] | R(2+1)D-18 (33.3M) | IG65M | 21 | VA | | 94.2 | | 67.4 | | | |
| ELo [64] | R(2+1)D-50 (46.9M) | YT8M | 13 | VFA | | 93.8 | 64.5 | 67.4 | | | |
| AVID [55] | R(2+1)D-50 (46.9M) | AS | 1 | VA | | 91.5 | | 64.7 | **89.2** | | |
| GDT [62] | R(2+1)D-18 (33.3M) | AS | 1 | VA | | 92.5 | | 66.1 | 88.5 | | |
| GDT [62] | R(2+1)D-18 (33.3M) | IG65M | 21 | VA | | **95.2** | | 72.8 | | | |
| VA only (ours) | R(2+1)D-18 (33.3M) | AS | 1 | VA | 83.9 | 91.5 | 60.0 | 70.1 | 85.6 | 29.7 | 55.5 |
| VA only (ours) | S3D-G (9.1M) | AS | 1 | VA | 84.7 | 90.1 | 60.4 | 68.2 | 86.1 | 29.7 | 59.8 |
| VA only (ours) | S3D-G (9.1M) | AS+HT | 16 | VA | 86.2 | 91.1 | 61.5 | 68.3 | 87.2 | 30.6 | 59.8 |
| MMV FAC (ours) | S3D-G (9.1M) | AS+HT | 16 | VAT | 89.6 | 92.5 | 62.6 | 69.6 | 87.7 | 30.3 | 68.0 |
| MMV FAC (ours) | TSM-50 (23.5M) | AS+HT | 16 | VAT | 91.5 | 94.9 | 66.7 | 73.2 | 86.4 | 30.6 | 67.8 |
| MMV FAC (ours) | TSM-50x2 (93.9M) | AS+HT | 16 | VAT | **91.8** | 95.2 | **67.1** | **75.0** | 88.9 | **30.9** | 70.5 |
| Supervised [19, 40, 64, 71, 87] | | | | | | 96.8 | 71.5 | 75.9 | 86.5[†] | 43.9 | 81.8 |

Table 3: **Image classification results on PASCAL and ImageNet.** "V→I" denotes the image handling strategy for the video networks: **n**aive **def**lation (no training of $\gamma$ and $\beta$), **def**lation (proposed), and **i**nput-**inf**lation (video net ingesting 32-frame static videos).

| Method | V→I | Train data | PASCAL (mAP) | ImageNet (top1) | ImageNet (top5) |
| --- | --- | --- | --- | --- | --- |
| Supervised S3D-G | def | Kinetics | 67.9 | 42.8 | 68.0 |
| MMV S3D-G | n-def | AS+HT | 41.8 | 20.7 | 40.5 |
| MMV S3D-G | def | AS+HT | 71.4 | 45.2 | 71.3 |
| MMV S3D-G | i-inf | AS+HT | 72.1 | 46.7 | 72.5 |
| Supervised TSM | def | Kinetics | 66.9 | 43.4 | 68.3 |
| MMV TSM | n-def | AS+HT | 34.4 | 10.9 | 24.6 |
| MMV TSM | def | AS+HT | 74.8 | 50.4 | 76.0 |
| MMV TSM | i-inf | AS+HT | 75.7 | 51.5 | 77.3 |
| Supervised TSMx2 | def | Kinetics | 66.9 | 47.8 | 72.7 |
| MMV TSMx2 | n-def | AS+HT | 45.6 | 20.3 | 39.9 |
| MMV TSMx2 | def | AS+HT | 77.4 | 56.6 | 81.4 |
| MMV TSMx2 | i-inf | AS+HT | 77.4 | 57.4 | 81.7 |
| SimCLR [13] ResNet50 | / | ImageNet | 80.5 | 69.3 | 89.0 |
| SimCLR [13] ResNet50x2 | / | ImageNet | / | 74.2 | 92.0 |
| SimCLR [13] ResNet50x4 | / | ImageNet | 84.2 | 76.5 | 93.2 |

# 5 Conclusion

In this paper we have explored how to train *versatile* networks for *vision*, *audio* and *language* in a self-supervised manner. Our method is simple yet it matches or exceeds the state-of-the-art for *action* and *audio* classification on five challenging benchmarks: HMDB51, UCF101, Kinetics600, ESC-50 and AudioSet. We encourage future work to use Kinetics600 and AudioSet that are larger scale downstream tasks and hence can better capture the progress of self-supervised methods. Our network can also be used for zero-shot text-to-video retrieval. Our *deflation* process shows how to train on videos to obtain representation for still images. Given the sheer number of videos available for self-supervised training on the web, we believe this is a more natural route to transfer which we hope will be pursued in the future.

# 6 Broader impact

**Potential benefits.** Our method can enable a better user experience when searching for visual or audio content on the web since we can index that type of media based on our learned multimodal embeddings. More broadly, learning video representations without labels in such a self-supervised manner greatly increases the scale at which we can train models, to the extent of leveraging any available collection of web video data. This enables capturing a more representative view of the overall distribution of web content as opposed to smaller scale curated datasets such as Kinetics. We believe this can be an important factor in designing methods that better understand whether or not a given content is safe (*e.g.* to filter out violent or undesired web content) thanks to the better coverage of the overall distribution.

**Potential risks.** Every method that learns from data, self-supervised methods even more deeply so, brings the risk of learning biases and perpetuating them in the form of decisions. We encourage the deployment of our method to be done with careful consideration of the consequences from any potential underlying biases in the data.

# 7 Funding

This work was funded by DeepMind.

# Acknowledgement

The authors would like to thank Antoine Miech, Yusuf Aytar and Karen Simonyan for fruitful discussions as well as Luyu Wang and Elena Buchatskaya for help on the evaluation benchmarks. We also want to thank our NeurIPS reviewers and metareviewer for great feedback on the paper.

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
