[Reviews · NeurIPS 2020]

Review 1

Summary and Contributions: This paper proposes MultiModal Versatile (MMV) networks that learns representations from vision, audio and text in a self-supervised manner. The main contributions include: + A hierarchical fine and coarse spaces approach (FAC) to configure the relationships between different modalities, which specifically tackles the issue that audio and vision modalities are more fine grained than the textual modality. + A network deflation mechanism to transfer the learned video representations to images. + Solid experiments and state-of-the-art performance evaluated on action recognition (UCF101/HMDB datasets), audio classification (ESC-50), zero-shot text-to-video retrieval (MSRVTT and YouCook2) and image classification (PASCAL VOC and ImageNet).

Strengths: + I'm convinced about the direction of multimodal self-supervised learning being a promising route to good representations. + The main technical contributions (FAC and network deflation) are sound. + Though simple, the techniques proposed in this paper (in particular FAC) makes sense and look novel to me. + The experiment protocols look sensible to me and the results obtained with the proposed method are convincing.

Weaknesses: - In Table 1a, I'm a bit surprised by the significant gains when comparing VAT to VA for UCF and HMDB, as I'm not sure why texts obtained from ASR play such an important role for datasets like UCF and HMDB? It would be great if the authors can provide more insights here. - The main difference between the proposed deflation mechanism and a "naive deflation" solution lies in the fact that the former re-tunes the parameters of batch norms in the network using a "L1 loss between the output of the original video network when presented with single-image static-videos, and the output of the deflated network for the same images". If re-tuning is indeed the important factor here, I think a more straightforward baseline is missing: the authors could fine-tune on PASCAL VOC with everything in the backbone frozen except gamma and beta in batchnorms. This way, one doesn't need to have the extra L1 loss to adjust the distribution for batchnorms. - If possible, I am curious how much better could the proposed method do when trained on even larger datasets (e.g., AS+HT+IG65M). This would help us understand the limits of the proposed multimodal self-supervised learning approach. - L173, what exactly constitutes "N(x)"? Are all N-1 negative pairs utilized? Could the authors elaborate?

Correctness: Yes, both the approach and experiment protocols look reasonable to me.

Clarity: It's very well-written and easy to follow.

Relation to Prior Work: Yes, the authors did a good job discussing related work. To provide a more comprehensive view on current state-of-the-art, I encourage the authors to add the following papers to Table 2, "Audiovisual SlowFast Networks for Video Recognition" and "Audio-visual instance discrimination with cross-modal agreement".

Reproducibility: Yes

Additional Feedback: POST REBUTTAL Although in certain cases it's hard to have exact apples-to-apples comparison (as the authors highlighted some practical difficulties in their rebuttal), I feel the results they provide does give me confidence in the story they're trying to sell ("find and coarse" multimodal learning). Thus, I will keep my rating as in the initial review. ------------------------------------------------------------------------------ Please see my comments in above sections.


Review 2

Summary and Contributions: In this work, the authors learn representations using self-supervision by leveraging three modalities naturally present in videos: vision, audio and language. A multimodal versatile network is proposed. Such a network that can ingest multiple modalities and whose representations enable downstream tasks in multiple modalities. The authors demonstrate the proposed approach by training the networks on large collections of unlabelled video data and applying to video, video-text, image and audio tasks.

Strengths: This paper is well written. The introduction very clearly presents the motivation of building such a multimodal versatile network. The representation of the idea, technical detail are also easy to follow/. I also like the idea that learning the multimodal embeddings in the fine-and-course manner. It is different with most of previous works that focus on learning a common space.

Weaknesses: 1. I am concerning about the novelty of the propose approach. Though the motivation of the idea and is to design a fine-and-course learning manner. However, the loss function is actually a combination of two contrastive loss, i.e. an audio-video contrast and a text-video contrast. Such a training objective is kind of regular in itself, unless the authors can clarify something novel despite that. The only difference is the model are trained on 3 different modalities, but I did not see any particular designing in dealing with the moltimoal data and the combination of the multimodal contrast loss. 2. I am concerning about the mismatch with the authors claim and their approach. The authors claim that they aims to learn the multimodal embeddings in the fine-and-course manner, which is different with previous disjoint and shared embedding space. However, the objective design is actually two contrastive loss, i.e. L_{a,v} and L{t,v}. In other words, it uses videos as an anchor to contrast with audio and text, respectively. However, in this way, it seems like no different with learning a the disjoint audio-video space and text-video space. With no clarification on this, the claim can not be validated. 3. Only a few SOTA are compared . There are more approaches and benchmarks are needed to be compared with. For example, on activity classification and audio event classification: [1] Audio-Visual Instance Discrimination with Cross-Modal Agreement [2] LEARNING VIDEO REPRESENTATIONS USING CONTRASTIVE BIDIRECTIONAL TRANSFORMER 4. the current performance are not outperform all SOTA, and the comparison is not fair. When all trained on AudioSet, XDC achieves 91.2, AVID [1] achieves 91.5 on UCF101, which are all outperform more than 1% on this work (90.1). On ESC, AVID achieves 89.2 while this work is 86.1, there is more than 3% gap. The best performance of this paper comes from training on a combination of AudioSet and HowTo100M that are too much larger than the other compared approaches. Considering the model is trained on much large dataset and also incorporating more modalities, the current performance is NOT surprisingly good. 5. The authors claim that, one of the benefits of using fine-and-course design is that, the information/knowledge embedded in specific modality can be easily transfer/translated to another, e.g. gv->gva->gvat. Thus, I am curious to see some experiments and analysis in the transferability of the learned network. Howeve, only video-text retrieval can valid the network learns some alignments between video and text. However, one may suspect that, it is learned from the video-text contrastive loss. Considering that, during training the network see the connection of audio-video and video-text, so it is natural the model can have the transfer ability between audio-video and video-text. But in order to prove that the whole network is transferable, the author also need to prove that the information/knowledge can be transfer/translated through audio-text. However, I did not see clear clarification in the approach part, also there is no experiments to empirically valid this. 6. There lack of related works on learning multimodal data (audio, video, text) [3] Unpaired Image-to-Speech Synthesis with Multimodal Information Bottleneck [4] One Model To Learn Them All 7. lack of enough implementation details, and I did not see the code.

Correctness: I am concerning about the mismatch with the authors claim and their approach. The authors claim that they aims to learn the multimodal embeddings in the fine-and-course manner, which is different with previous disjoint and shared embedding space. However, the objective design is actually two contrastive loss, i.e. L_{a,v} and L{t,v}. In other words, it uses videos as an anchor to contrast with audio and text, respectively. However, in this way, it seems like no different with learning a the disjoint audio-video space and text-video space. With no clarification on this, the claim can not be validated. The authors claim that, one of the benefits of using fine-and-course design is that, the information/knowledge embedded in specific modality can be easily transfer/translated to another, e.g. g_v->g_va->g_vat. Thus, I am curious to see some experiments and analysis in the transferability of the learned network. Howeve, only video-text retrieval can valid the network learns some alignments between video and text. However, one may suspect that, it is learned from the video-text contrastive loss. Considering that, during training the network see the connection of audio-video and video-text, so it is natural the model can have the transfer ability between audio-video and video-text. But in order to prove that the whole network is transferable, the author also need to prove that the information/knowledge can be transfer/translated through audio-text. However, I did not see clear clarification in the approach part, also there is no experiments to empirically valid this.

Clarity: Yes. This paper is well written. The introduction very clearly presents the motivation of building such a multimodal versatile network. The representation of the idea, technical detail are also easy to follow/.

Relation to Prior Work: No. This paper lacks of some related works, especially for the multi-modality works ( 'vision, audio and language' ). Considering the one of the main technical contributions is learning through all of these three modalities, the current related work and discussion is apparently not satisfied. Besides, the authors claim that the main difference is to learn a fine-and-course space instead of previous 'shared space' and 'disjoint space'. However, from the approach part, I can not be convinced that the current objective design can achieve what the authors claimed. There is no clear discussion in differentiating the proposed approach with previous ones. [1] LEARNING VIDEO REPRESENTATIONS USING CONTRASTIVE BIDIRECTIONAL TRANSFORMER [2] Unpaired Image-to-Speech Synthesis with Multimodal Information Bottleneck [3] One Model To Learn Them All

Reproducibility: No

Additional Feedback: -------------------------POST REBUTTAL------------------------------------------------------------ Thanks for all your discussion. Yes, I agree that the incomplete comparison should NOT be the key factor that affects contribution of this paper. The authors' feedback also addressed my concern in it, so I am fine for Evaluation part. For my concerns about the technical novelty, the authors did address part of my concerns by clarifying the main contribution is the architecture design rather than the loss function. However, the designing does not fully convinced me. Though I like the idea of 'fine-to-course', but the architecture design and training loss do not achieve this motivation in a novel way. Considering these, I would raise my score to 5, but I will be fine if this paper gets in. Because it is well written and the 'fine-to-course' concepts is new and interesting.


Review 3

Summary and Contributions: The author presents a self-supervised representation learning framework for videos that contain multimodal streams of information. In particular, two components are emphasized in the paper: Fine and Coarse Spaces (FAC) and the deflation method. Empirical experiments manifest the effectiveness of the approach. ---------------------- i will remain my rating and still think the paper is worthy of acceptance. I will maintain my score 6.

Strengths: The experimental results are promising. The paper is easy to follow and understand. Nonetheless, many of the details are hidden in the main text.

Weaknesses: (Minor) The paper fails to discuss a series of related work on human multimodal language. Details will be provided later. (Minor-Major) In lines 174-176, the author claims that the audio is perfectly aligned with the visual source, which may not be valid. Different sampling rates or various angles of the visual scenes may cause misalignment between visual and audio signals. (Minor-Major) The author should discuss more on how the positives are chosen for the text. Details will be provided later. (Minor-Major) The author should elaborate more on the deflation process in Section 3.2. (Minor) The experimental section is a bit hard to follow. Nonetheless, I believe these positive results.

Correctness: The author states the visual and audio is always aligned in a given video, which may fail for most of the cases.

Clarity: 1. It is unclear how the positive text signals are sampled. For instance, lines 180-181 is particularly vague. Also, what is P(x) here? 2. I can't grasp how important is the deflation introduced in the paper. Perhaps more discussions or better explanations can be made in section 3.2.

Relation to Prior Work: There are a series of important work [1,2,3] on multimodal video should be discussed. These works also consider vision, text, and audio signals in the video. [1] Tensor Fusion Network for Multimodal Sentiment Analysis, Zadeh et al., EMNLP'17. [2] Multi-attention Recurrent Network for Human Communication Comprehension, Zadeh et al., AAAI'18. [3] Multimodal Transformer for Unaligned Multimodal Language Sequences, Tsai et al., ACL'19.

Reproducibility: No

Additional Feedback: n/a


Review 4

Summary and Contributions: This paper proposes a new approach to multi-modal self-supervised representation learning. This new approach (called MMV) is a contrastive learning-based approach that learns from three modalities: visual, audio, and text. MMV maps the three modalities into a fine-and-coarse (FAC) feature space. The paper evaluates the performance on multiple downstream tasks and achieves good results.

Strengths: - The paper is evaluated on many downstream tasks - To the best of the reviewer's knowledge, this is the first self-supervised approach to work on the three modalities: vision, audio, and text - The ablation studies are well-conducted

Weaknesses: The main weakness of this work is the lack of direct comparisons with previous works. The authors claim that they outperform the state-of-the-art on UCF, HMDB, and ESC, but fail to compare with other approaches under the same backbone architecture and the same pertaining dataset. For example, the following concerns arise from table 1: - Is MMV with TSM-50 better than ELo because MMV has a better architecture and was pretrained on a larger dataset? What is the performance of ELo pretrained on AudioSet+HowTo100M using TSM-50? Or What is the performance of MMV using R(2+1)D-50 and pretrained on YouTube8M? - XDC outperformed its R(2+1)D-18 fully-supervised pertaining baseline, but MMV did not exceed its fully-supervised pertaining baseline. On the other hand, MMV shows better performance than XDC. Can the authors explain why this is the case? Is MMV better than XDC because it uses a better backbone architecture? - MIL-NCE uses the same S3D backbone as MMV, but MIL-NCE is only pretrained on HowTo100M. Could the additional AudioSet pertaining be the reason why MMV outperforms MIL-NCE by 1.3% on UCF? What is the performance of MMV trained only on HowTo100M? - AVTS uses MC3 architecture, which has an inferior performance to TSM-50 on action recognition tasks. Could this be a factor why AVTS shows worse performance on UCF compared to MMV? What is the performance of AVTS using TSM-50? Or what is the performance of MMV using MC3? Overall, the lack of a fair direct comparison with any of the previous approaches undermines the claim that MMV is a better representation-learning model. The authors should compare with at least one method under the same settings (the same architecture and the same pretraining dataset).

Correctness: The claim that MMV is better than previous works is not backed by fair direct comparisons using the same backbone architecture and pretraining dataset. Refer to the "Weekeness" section for more details.

Clarity: The paper is well written and organized

Relation to Prior Work: yes

Reproducibility: Yes

Additional Feedback: **Post-rebuttal review updates:** After reading the authors' rebuttal, I have the following comments about the response to my concerns about the lack of direct comparisons with other works (i.e. on the same architecture and same pretraining dataset): Authors: "Comparison on equal grounds is a problem for all papers in this area" - Yes, that's sadly true - but that's why we need to find a way to fix it. It's unreasonable (and it is not what I asked for) to ask the authors to compare directly against all previous methods, but they should at least have one direct comparison using the same backbone and pretraining dataset. Otherwise, we cannot know if a new paper is actually better than others because of its specific approach or due to the use of advanced backbones and large datasets (two factors that are not novel by themselves). Authors: "We did another comparison to XDC (R2, R4) by running our VA model on the same data (AudioSet) and backbone (R(2+1)D-18) ... Note that R(2+1)D-18 actually outperforms S3D " - The authors providing this experiment is exactly what I was asking for to convince a reader about the merits of this paper. Now we can directly compare to another method (XDC), and now we can fairly say MMV is a better method. That being said, I'm kinda surprised by the fact that using R(2+1)D-18 gave MMV better results than using S3D. In the supervised action recognition on Kinetics, S3D outperforms R(2+1)D-34 (note this is the 34-layer variant), let alone the smaller and weaker R(2+1)D-18 model. I'm not sure what might cause this mismatch, but I will take the authors' findings as correct. However, the authors should discuss this surprising mismatch. Authors: "XDC beats their own fully supervised baseline but we report a stronger and more meaningful quantity – the best externally published performance for supervised transfer" - I'm afraid this is a false statement. The reported fully-supervised results in Table 2 are actually the performance of S3D itself [55] (which is the fully-supervised baseline for MMV). Specifically, the 96.8 on UCF and 75.9 on HMDB are taking directly from Table 5 of the S3D paper [55]. So, I still stand by my concern here. MMV appears to underperform its fully-supervised baseline and at the same time outperforms XDC, which outperforms its fully-supervised baseline. The authors' response is misleading and does not address my concern. That being said, this does not mean that MMV is worse because it failed to outperform its fully-supervised baseline. But the author should have a convincing argument as to why it does not. I like this paper and I think it's the first SSL approach to work on the three modalities: vision, audio, and text. The authors' response addressed part of my concerns by directly comparing to one method. I'm leaning towards changing my rating from 4 to 5 as I still think the authors failed to address my remaining concerns.

[Author Response · NeurIPS 2020]

We thank reviewers for their thoughtful feedback.

**Additional references (R1, R2, R3).** We reported in Table 2 the best existing self-supervised baselines at the date of
submission (including unpublished methods such as XDC). We outperform all the reviewers' additional references that
target unsupervised learning (R2's [1,2]), apart from results on a single benchmark (ESC-50) for R2's [1] which we
indeed miss and will add (it appeared on arXiv one month before the deadline and is unpublished). Other references are
relevant but do not address the problem of unsupervised representation learning (e.g. "One model to learn them all" is
about supervised learning, "Multimodal transformer for unaligned multimodal language sequences" uses supervised
features...). We will add all mentioned references but we stress that they do not hurt our novelty claim nor our results.

**Fair comparison to previous work (R2, R4).** Comparison on equal grounds is a problem for all papers in this area,
and we try to be fair: (1) The **comparison with MIL-NCE (R4)** is performed in Table 1(a) since MIL-NCE is strictly
equivalent to our VT only architecture. With the exact same training setup and backbone, our FAC outperforms
MIL-NCE by 2% in both UCF and HMDB, while enabling a task (ESC-50) that would not be possible with MIL-NCE.
(2) The point of the paper is to demonstrate that **more modalities help (R2)**. Note that we outperform ELo which
uses an additional modality that we don't (flow, which is really important for action recognition). (3) It is difficult
to compare our method on the **same data and architectures** as (i) FAC requires text and only HowTo has it (only
MIL-NCE uses this dataset, and we compare to this), (ii) IG65M is not public (**R1**), (iii) AVTS, XDC and ELo haven't
released code, and (iv) other works use a variety of architectures. We beat XDC despite using less data (16 vs 21 years)
and fewer parameters (24 vs 33M), and ELo despite using a $2\times$ smaller model. (4) We did **another comparison to**
**XDC (R2, R4)** by running our VA model on the same data (AudioSet) and backbone (R(2+1)D-18) and outperform
them: 91.5 (vs. 91.2) on UCF and 70.1 (vs. 61.0) on HMDB (also matching R2's [1] on UCF and beating on HMDB on
the same data). Note that R(2+1)D-18 actually outperforms S3D (used in our submission) so MMV does not simply
beat XDC due to a better backbone (**R4**). (5) **XDC beats their own fully supervised baseline** but we report a stronger
and more meaningful quantity – the best externally published performance for supervised transfer. Finally, as detailed
next, we would like to stress that our good performance (e.g. a significant boost of **5.5** point on HMDB) is not the only
contribution of the paper.

**Novelty, contributions and claims (R2).** We agree that our novelty does not lie in the loss which is indeed not novel.
The loss is not the only means to induce different structures on the embeddings, instead, we achieve that through
architecture design. For example, the FAC design allows us to navigate from the video-audio space (fine) to the
video-text space (coarse) (property (iii) of the MMV L28), which is not possible with the disjoint design. We validate
that claim in the supplemental video as highlighted in L313 through qualitative audio-to-text retrieval (we are not aware
of standard benchmarks for quantitative audio-text retrieval evaluation). This shows that we can influence the structure
of the embedding through architecture design and not only with losses. Furthermore, we show that FAC performs better
than the other considered designs in Table 1(b) on 3 out of 5 benchmarks. In addition, to the best of our knowledge
(acknowledged by R1 and R4), this is the first work to jointly learn from video, audio and text in a self-supervised
fashion. In the paper, we explore how to do that well at scale, propose various embedding strategies, and demonstrate
state-of-the-art performance on challenging downstream tasks which we deem to be an impactful contribution.

**Importance of the deflation contribution (R3).** We believe the deflation technique is an important contribution of
the paper as it allows video-trained models to do inference efficiently with image inputs. In particular, we show that the
image classification performance when using the deflated model is similar to using the original model with an inflated
input. In addition, we believe to be the first work to consider learning first from video to transfer to images and show
strong performance on two image benchmarks. We will clarify the impact of the deflation contribution in the paper.

**Baseline for deflation (R1).** R1's proposed method is indeed a valid idea which we expect to perform on par or
better than our approach. However, this method effectively does partial finetuning, while we instead focused on linear
evaluation with frozen networks since it enables fast off-the-shelf evaluation on unseen image dataset and tasks.

**Positives and negatives for NCE (R1, R3).** For the text positive (L180), we strictly follow MIL-NCE [41] and refer
to this paper for details. In L173, we use all negatives coming from other elements in the batch. In total there are
$2 \times (N-1)$ negative pairs ($N-1$ audio negatives for the video and $N-1$ video negatives for the audio).

**Model and code release (R2, R3).** We will release our pretrained models as well as the training scripts.

**Perfect audio-visual alignment (R3).** We agree there is no perfect alignment, but the point of L174 was to emphasise
that text is less aligned with the visual content than audio in general. Note that all competing audio-visual learning
approaches ignore the occasional misalignment and, like us, observe that learning is possible despite it. We will clarify.

**ASR for action recognition (R1).** Text from ASR provides semantic information about the visual content of the
videos for objects (e.g. tomato) or actions (e.g. cut). Interestingly even though there exists a domain gap between the

[Meta-Review · NeurIPS 2020]

This paper received mixed reviews: R1 recommends clear accept (score 8), R3 recommends weak accept (score 6), and R2 & R4 recommends weak reject (score 5). All four reviewers agreed that the tri-modal learning formulation (vision, audio and language) is interesting and the FAC (fine and coarse) approach is conceptually novel. R1 & R3 acknowledged that the experiments are extensive and convincing, and R4 noted that the ablation studies are well conducted. EVALUATION -- R2 & R4 shared a concern that some of the comparisons with prior work are problematic because of different experimental protocols (backbone architectures and datasets used for self-supervised pretraining). The rebuttal addressed this by reporting new experimental results obtained using the same backbone/dataset with the baselines. During the discussion phase, R2 acknowledged that the new results addressed the reviewer's concern. R4 also acknowledged the same, but seem to remain lukewarm because of one potentially inaccurate statement (see R4's review); it would be great if the authors could clarify this carefully in the paper and discuss what the implications are. NOVELTY -- R2 raised a concern about technical novelty. Although FAC is conceptually novel, the actual implementation of the idea is based on existing networks and loss formulations, and thus the proposed approach does not achieve the FAC motivation in a novel way. The authors agreed to this point but argued that their claimed novelty is in the design of FAC embedding process. R2 didn't seem to be convinced by this argument despite the rebuttal. I think R2 made a valid criticism here and agree that the concern isn't properly addressed. I would also add that this paper does not provide convincing evidence showing the superiority of FAC over the other alternatives (Shared and Disjoint). This can be shown in Table 1(b): FAC performs better on UCF/HMDB/MSRVTT, but Shared performs better on ESC50 and Disjoint performs better on YouCook2 (and all the performance differences are all marginal within about 2% differences). Furthermore, the experimental setup makes it difficult to draw meaningful conclusions. The results show inconsistent results across different downstream tasks, including action classification (UCF/HMDB), sound classification (ESC50), and video/language understanding (MSRVTT/YouCook2). What conclusions can we draw from such result? When should we use FAC instead of Shared/Disjoint approaches? Should we choose FAC only for action recognition, or for all scenarios? When should we learn from all three modalities? Is learning from all three modalities always a better idea or should we be more judicious about the choice of modalities depending on possible downstream scenarios? Why would bringing sound & text help solve action recognition, which is inherently vision-centric? (R1 asked a similar question; the rebuttal tried to answer this but unfortunately the text was cut due to space limit) These questions are hard questions to answer thoroughly, but I think the authors could've run controlled experiments to provide better insights and generalizable conclusions. That said, despite my added criticisms above, I am convinced about the general direction of multimodal representation learning. This paper provides good empirical performances on several challenging benchmarks; I especially liked the results on MSRVTT/YouCook2 which have not been used frequently in the self-supervised video representation literature (I personally think we are reaching the limits of UCF/HMDB as testbeds for self-supervised learning). The scale of the experiments is also impressive (HowTo100M+AudioSet pretraining, evaluation on UCF/HMDB/ESC50/MSRVTT/YouCook2). As R4 mentioned, various ablation studies are well-conducted as well. These will set the bar higher for future research in this direction. Given this, I think this paper is worth acceptance. It is imperative that the authors will include the new results presented in the rebuttal, and discuss various references pointed out by the reviewers.